# Dental Treatment under General Anesthesia in Pre-School Children and Schoolchildren with Special Healthcare Needs: A Comparative Retrospective Study

**DOI:** 10.3390/jcm11092613

**Published:** 2022-05-06

**Authors:** Nelly Schulz-Weidner, Maximiliane Amelie Schlenz, Linda Giuliana Jung, Constanze Friederike Uebereck, Agnes Nehls, Norbert Krämer

**Affiliations:** 1Department of Paediatric Dentistry, Dental Clinic, Justus Liebig University, Schlangenzahl 14, 35392 Giessen, Germany; linda.g.jung@dentist.med.uni-giessen.de (L.G.J.); constanze.uebereck@dentist.med.uni-giessen.de (C.F.U.); norbert.kraemer@dentist.med.uni-giessen.de (N.K.); 2Department of Prosthodontics, Dental Clinic, Justus Liebig University, Schlangenzahl 14, 35392 Giessen, Germany; maximiliane.a.schlenz@dentist.med.uni-giessen.de; 3Dental Practice, Kinderzahnärzte Am Ostpark MVZ Gmbh, Albert-Schweitzer-Str. 66, 81735 München, Germany; nehls@muenchen-kinderzahnarzt.de

**Keywords:** pre-school children, early childhood caries, special healthcare needs, risk factors, pediatric dentistry, dental treatment, general anesthesia, caries prevention, oral health

## Abstract

The aim of this retrospective study was to evaluate the dental treatments received under general anesthesia (GA) in pre-school children and school children with special healthcare needs (n = 263) compared with healthy controls (n = 62). In order to analyze the impact of pre-existing cofactors on oral health, children were divided into the following groups: heart disease, cancer, syndromic disease, and other diseases as well as in- and outpatient healthy children. Caries experience (dmf-t/DMF-T) before GA (impulse 1), waiting time, and dental treatment under GA (impulse 2) were determined. Pre-school children showed a higher caries experience (mean ± standard deviation; 8.3 ± 4.8) compared to schoolchildren (6.9 ± 4.3). Predominantly, early childhood caries (ECC) of type 1 were detected. From all groups with special healthcare needs, children with cancer revealed the highest Significant Caries Index (17.0 ± 2.0), followed by other diseases (14.6 ± 2.6), syndromic disease (14.3 ± 1.8), and heart disease (13.8 ± 2.7). Overall, 2607 dental procedures were performed under GA with a mean of 8.0 ± 6.5 dental measures per child. Within the limitations of this study, the data exhibited independent of pre-existing cofactors and age high caries risk in all patient groups showing a high need for treatment.

## 1. Introduction

Several studies described the caries experience in representative samples of 6-year-old and 12-year-old children since 1993 [1,2,3,4]. Furthermore, data of 3-year-olds are available [5]. Especially for the group of 3-year-olds, the insufficient treatment of ECC could be shown [5]. Beyond that, there is only rare data on the caries experience of children with pre-existing cofactors, so-called special healthcare needs. Thus studies show children with disabilities [6,7], cancer [8], or heart disease [9] are more likely to have more dental caries compared to healthy children. Additionally, they suffer from untreated dental caries, lesions, infections, gingivitis, periodontitis, and other problems [9,10,11].

According to the available studies, children with special healthcare needs show a higher caries experience and severity in comparison to healthy ones. This is the reason why dental treatment is mostly carried out under general anesthesia (GA). Indices such as the Significant Caries Index (SiC) and the Caries Restoration Index (CRI) are helpful tools to be able to assess caries experiences and treatment needs in groups with a high caries incidence [12,13]. However, there is a lack of data comparing different groups of children at different ages and pre-existing cofactors suffering from severe caries, which make dental treatment under general anesthesia indispensable. To the best of the authors’ knowledge, this is the first comparative study regarding caries experience and following dental treatment under general anesthesia in children with special healthcare needs according to age.

The need for timely caries therapy in the primary dentition is beyond question. Without dental treatment, dental caries can cause pain and infection [14]. Moreover, the disease can affect general health [15], especially in children with special healthcare needs.

However, the medical situation may overtax pre-school children because of the lower developmental stage [16,17]. For children with special healthcare needs the often high need and associated length for treatment may be go beyond the possibilities of chairside treatment. In these cases, treatment under GA offers sufficient dental comprehensive conservative and surgical rehabilitation.

Until a few years ago, dental rehabilitation under GA was mainly in the hands of, on one hand, special clinical centers for dentistry and oral and maxillofacial medicine and, on the other hand, in specialist practices for oral and maxillofacial surgery with inpatient care options in regional hospitals [18]. Due to the specialists focus on the dental side and the integration of outpatient specialists in anesthesia, the number of outpatient dental treatments has increased in recent years. Both Savanheimo and Vehkalahti as well as Ramazani demonstrated that an increasing number of patients, with age-appropriate inadequate or exhausted cooperation, were treated in a specialized pediatric practice in outpatient dental treatment under GA [19,20]. Not only pre-school children but also older school children with limited or missing cooperation and, moreover, patients with reduced general health may need this special approach [17]. Due to the possible dental complications and the associated risk of anesthesia, particularly, very young children and the above mentioned special needs patients are usually rehabilitated in an inpatient medical setting either in university dental clinics or special practices with inpatient medical facilities.

In our Department of Paediatric Dentistry in Giessen (Justus Liebig University Giessen (JLU), Giessen, Germany) children are regularly treated in inpatient admission in cooperation with the Paediatric Center (Medical Centre of Paediatrics, JLU Giessen, Giessen, Germany) under GA. A major proportion of the children are younger than six years, and they suffer from early childhood caries (ECC). Furthermore, different pre-existing general health care problems are often obvious.

The characteristics of the mentioned necessary comprehensive dental treatment under GA have already been reported in several studies with a focus on outpatient dental treatment [21,22,23,24,25]. However, in the inpatient group, there is a lack of significant data available. Regardless, according to this rare data, oral rehabilitation under anesthesia is generally considered a suitable treatment option, especially for children requiring special care [26,27]. Furthermore, children with a disability or a serious general medical background are treated following this schedule. In this context, there is a necessity to mention that even dental examinations for prevention and prophylaxis and the pre-assessment of the planned dental therapy may be very demanding and time-consuming due to overload of children and caregivers or to missing cooperation or unwillingness to cooperate [28]. Furthermore, due to limited possibilities, waiting time before dental intervention could be long [29].

Regarding patients with special healthcare needs, dental treatment is normally scheduled as one-step therapy resulting in a more extensive path of therapy to prevent secondary treatments. Furthermore, these children are often unable to maintain the necessary oral hygiene, so teeth are extracted more frequently in comparison with a conservative treatment approach probably ending in failure [30].

The aim of our present study was to evaluate the dental treatments received under GA in pre-school children and school children with special healthcare needs compared to healthy controls.

In detail, caries experience (dmf-t/DMF-T) before dental treatment under GA (impulse 1) were determined. Furthermore, waiting time for and dental treatment under GA (impulse 2) were evaluated.

The main null hypotheses for this study were formulated:(1)There is no association between caries experience (dmf-t/DMF-T, SiC and CRI) and pre-existing cofactors.(2)There is no difference in waiting time regarding pre-existing cofactor.(3)There is no difference in terms of dental treatment in all groups.

## 2. Materials and Methods

### 2.1. Subjects and Setting

A cross-sectional study with retrospective data collection was designed. The patient population comprised all children between 0 and 18 years who underwent comprehensive dental treatment under GA in the period from 2013 to 2019 at the Department of Paediatric Dentistry (JLU Giessen, Giessen, Germany) as well at a specialized practice in Munich. The standard written informed consent for dental treatment under general anesthesia was obtained from caregivers before treatment. The exclusion criteria were incompleteness of dental records and missing written informed consent.

The data research focused on two following main patient groups: children with general diseases and healthy children. The latter included children with a healthy general condition and/or without a significant handicap (maximum ASA classification I [31]). Furthermore, for comparing characteristic treatments in medical in- and outpatient conditions a group of healthy children was also included from the Munich practice (healthy control group). Additionally, according to the ECC definition, groups were divided into two age groups, younger (pre-school children), and older (school children) than six years [32].

The study was realized in accordance with the guidelines of the Declaration of Helsinki, and it was approved by the local ethics committee of the Department of Medicine, JLU Giessen (AZ 104/19).

### 2.2. Data Collection

The files, applied to the study addressed here, were carefully filtered after a manual archive search and checked with regard to the inclusion and the exclusion criteria. All children treated under GA were registered in a database. All existing files of the Department of Paediatric Dentistry in Giessen were searched for the main target parameters of the present study (caries experience, waiting time, and performed dental procedures under GA). The patient information was compared with this database, and missing information was added manually. To record the child’s general and special medical history, the contents of the patient records and doctor’s letters were evaluated. Each patient was labeled by a consecutive identification number.

For general history, the patients were divided into six patient groups: Healthy in- and outpatients as mentioned above and according to the “International Statistical Classification of Diseases and Related Health Problems-ICD-10” [33] into four further patient groups, as follows: heart disease, cancer, syndromic disease, and other diseases.

According to the dental examination, the dmf-t/DMF-T index was obtained for evaluation of caries experience. A tooth was considered as decayed (d) if a carious lesion (without consideration of initial carious lesions) was found, as missing (m) if the reason for loosening the tooth was caries, and filled (f) if there was a restoration [34]. Furthermore, the Significant Caries Index (SiC) demonstrating the mean dmf-t/DMF-T value of the one third with the highest caries level of all patients was analyzed [12,35]. The degree of caries restoration was an indicator of the need for caries treatment using Caries Restoration Index (CRI) [1,36,37]. Special attention was paid to ECC categorization [38]. The dmf-t was assigned to children with sole primary dentition, the dmf-t-/DMF-T to children presenting mixed dentition, and the DMF-T to patients already showing a permanent dentition.

The scope of the retrospective data collection comprised two impulses each. Impulse 1 corresponded to the initial examination (pre-operative assessment) of the child. At this appointment, administrative and clinical details were recorded. Different pediatric dentists trained at the same institution determined the need for GA and performed all dental procedures (N.S.W., C.F.U. and A.N.). As this study was retrospective and based on patient records, pre-treatment calibration was not possible. Impulse 2 included the treatment measures of GA rehabilitation, meaning the postoperative findings. The required restorative (fillings and/or stainless steel crowns) and/or surgical treatments for each patient were completed in a single session under GA under intubation. Teeth with deep carious lesions displaying pulp exposition received endodontic therapy depending on indication. Otherwise, the treatment of choice was extraction. The waiting time before treatment was also recorded. It was defined as the time between the initial examination and the surgery date.

Data about the follow-up evaluations could only be obtained in rare cases on account of our special children’s group with residences throughout Germany. This is the reason why follow-up visits were not compared.

### 2.3. Statistical Analysis

The data acquisition and the construction of graphics were accomplished with the DOS-based program dBASE IV (Borland, Austin, TX, USA) and Microsoft^®^ Excel (Office Version 2011, Microsoft Cooperation, Redmond, WA, USA). The statistical analysis of data was performed using the software package SPSS^®^ for Windows (version 26.0, IBM Corporation, Armonk, New York, NY, USA). Data were presented in the form of absolute and relative frequencies, and they were described by the mean value including their standard deviations. The differences between the groups were evaluated using an ANOVA test regarded as significant for *p* < 0.05.

## 3. Results

In total, 325 children aged between 0 and 18 years were enrolled. Based on age distribution, 165 children were pre-school children (mean age 4.0 ± 1.2, 75 girls, 87 boys, and 3 not stated), the other 160 school children were aged over 6 years (mean age 9.2 ± 2.7, 66 girls, 92 boys, and 2 not stated). Pre-existing general health problems were diagnosed in 126 pre-school children and in 137 school children.

### 3.1. Caries Experience

Figure 1 presents the comparison of caries experience (mean dmf-t/DMF-T) of pre-school children and school children for all treated groups at initial examination (impulse 1). Pre-school children showed a mean dmf-t value of 8.3 ± 4.8, whereas school children revealed a lower mean dmf-t/DMF-T of 6.9 ± 4.3. Overall, the dmf-t/DMF-T was always detected to be higher in the group of pre-school children compared with school children. In addition, the dmf-t value of the pre-school children with special healthcare needs was higher compared with healthy patients; this was more evenly distributed in the school children group, where the healthy inpatients in particular had the lowest mean value. Nonetheless, the results did not show a significantly higher caries incidence among the children with special healthcare needs, among all other groups and also in comparison with the in- and outpatient healthy children group (*p* > 0.05, ANOVA).

Table 1 details the clinical findings for all groups listed above demonstrating the need for caries treatment determined by the SiC Index [12] and the Caries Restoration Index [36,37]. Pre-school children revealed a higher SiC compared with school children. The highest need for treatment with a SiC Index of 17.0 ± 2.0 was demonstrated in the pre-school cancer group. The healthy in- and outpatient pre-school and school groups presented for each group comparable results. The CRI in pre-school children was calculated as low with the highest value in healthy inpatients, followed by the syndromic disease group. Syndromic school children revealed the highest CRI followed by children with cancer. All these data did not show a significant difference (*p* > 0.05).

The distribution of early childhood caries (ECC) is demonstrated in Figure 2. Regarding the 165 pre-school children, we could demonstrate 57% belonging to ECC Type 1, 26.2% to ECC-Type 2, and 12.2% belonging to ECC Type 3 [38]. Eight children were treated according to trauma (*p* > 0.05).

### 3.2. Dental Treatment under General Anesthesia

#### 3.2.1. Waiting Time

In 325 children, a total of 2607 procedures were performed under GA meaning 8.0 ± 6.5 procedures per child. The average waiting time was detected between 1.2 and 15 months depending on age and pre-existing cofactor. There was a higher waiting time in cases of children with special healthcare needs, with a peak of 6.3 months per child in the group of heart disease and syndromic disease in pre-school children and in the group of syndromic disease in school children. Children with cancer and healthy outpatients showed the lowest waiting times regardless of age (Table 2).

#### 3.2.2. Dental Procedures in GA Patients


Figure 3 presents the status of the dmf-t/DMF-T index at impulse 1 (IMP1) and impulse 2 (IMP2) for pre-school children. In all patient groups, fillings, and extractions were detected.

Regarding dental treatment in school children, extractions were less frequently found as dental measures for permanent teeth; more filling therapies were carried out (Figure 4).

Table 3 displays the dental procedures for all treated patient groups. Extractions in primary dentition and fillings in the permanent dentition were most common. Stainless steel crowns were only used in the primary dentition. In children with general diseases, the total number of teeth removed was 987, in comparison with 735 fillings, and 96 stainless steel crowns performed in primary dentition. Additionally, 334 fillings and 72 extractions had to be carried out in the permanent dentition. Regarding the healthy children, 268 fillings, and 73 extractions in primary dentition were performed (41 fillings and 1 extraction in permanent dentition). Endodontic procedures were not reported.

## 4. Discussion

According to the WHO criteria, we used the dmft-index for primary teeth and/or the DMFT index for permanent teeth (“decayed”/carious, “missing”/extracted, “filled”/filled “teeth”/teeth) for the recorded individual components of the index [34]. From the point of view of health services research, an individual definition of the dmf/DMF-index or a differentiation from the dmf/DMF components is necessary to provide the possibility for a comparative approach [13]. Since the prevalence of caries in children is not normally distributed, we used the SiC [12] and the Caries Restoration Index [37] in order to characterize the caries risk in the group of patients.

Our data exhibited a high caries experience in all age groups independent of coexisting diseases. Pre-school children showed a higher caries experiences compared with school children regardless of coexisting diseases. The mean dmf-t value of 8.3 revealed a very high caries experience in our pre-school study sample. This data showed approximately a dmf-t/DMF-T value four to five times higher with eight affected teeth compared with other studies [6,13,39]. These results are also not in accordance with Nies indicating lower caries risk according to age-group [40]. Additionally, our dmf-t/DMF-T value of school children, presenting a value of 6.9 was also detected to be significantly higher than in a study describing 12-year-old children [6]. In total, 263 of the 325 treated children were diagnosed with coexisting general health problems. Our results confirm poorer oral health in special healthcare needs patients showing higher caries experience compared with healthy children, which is in accordance with the literature [41,42,43,44,45,46,47,48,49]. Moreover, we were able to demonstrate a higher caries risk not only in the compromised children but also in healthy children. In particular, the increased SiC in our study confirms this aspect. Additionally, it was striking that the outpatient healthy children had higher dmf-t/DMF-T values than the inpatient healthy group. This may be explained by the lowered age in our inpatient group. From our practical experience, dental treatment in GA is only carried out from a certain weight and age, which is the reason for the younger healthy children treated in our pediatric dentistry.

In addition, in our study, a low level of CRI was observed among all children regardless of age. Untreated caries have already been described in special healthcare needs children [48]. Bird et al. pointed out a difference in dental care in children with disabilities compared with healthy children [49]. These results are similar to Pieper et al., demonstrating approximately half (47.4%) of the school-age children showing no intact fillings in carious teeth [50]. These results once again confirm the importance of consistent early dental care from infancy onward, regardless of coexisting concomitant factors. The fact that a medical condition is often associated with the inability to cooperate with chairside dental treatment requires that this special group of children has often be treated under GA [48].

The literature reports that a long waiting time can have an impact on the health of children. Delays in treatment can lead to deterioration of children’s teeth [51]. Another study showed that during the waiting period children needed painkillers and suffered from sleep disturbances and problems with chewing. Furthermore, many children took antibiotics [52]. The considerable waiting time of children with preexisting factors, and also of healthy children, shows the need for urgent dental treatment under general anesthesia and also the need for interdisciplinary care. Waiting times have to be shorter in future. Only rarely dentistry centers have the capacity for dental treatment of special healthcare needs patients resulting in high waiting times. This capacity-related waiting time shows that the need for dental treatment under GA outweighs the hesitancy, and it demonstrates the importance of pediatric postgraduate education as well the importance of prevention strategies to avoid caries in this vulnerable group.

Showing that children with special healthcare needs are a high caries risk group highlights the need for prevention in these patients. Our results indicate an obvious need for prophylactic programs including dental prevention measures accessible to all patients with and without special healthcare needs and with no age limit. In these cases, treatment under general anesthesia could be prevented. In addition, the success of dental interventions in GA depends on subsequent follow-up examinations [48], which require a close-meshed preventive recall program.

Despite contemporary preventive and minimally invasive treatment approaches in pediatric dentistry, tooth extraction is still a necessary treatment for advanced and multisurface caries with 1060 extractions vs. 808 fillings in the primary dentition. Our results are in accordance with other studies, showing higher percentages of extractions in primary dentition [53]. This can be explained, on the one hand, by more severe destruction of teeth, as underlined in our results in this special high caries risk group. Regarding special health care needs children, it underlines the characteristic therapy of these patients in extracting severely decayed teeth. Restorative measures performed under GA were higher in permanent dentition with a higher number of fillings performed per child. This data are well in accordance with the results of Rubin et al., demonstrating the necessity for restorative procedures [53] and similar to the experience of Campbell et al. showing dental treatment consisting of 8–9 teeth, including crowns and fillings [54]. It is also worth mentioning that the need for treatment in the initial examination usually exceeded the final therapy. In our opinion, this could be due to the fact that pre-assessment is extremely difficult to perform in this particular patient group and, therefore, the data are not recorded exactly.

In total, 26.2% (n = 43) of the children suffered from “nursing bottle syndrome”. This clinical manifestation was classified under ECC type II according to Wyne [38]. In a total of 12.2 % (n = 20) of the subjects, additional caries damage was observed on the lower teeth, corresponding to ECC type III. Müller-Lessmann et al. detected that 71.9% of the caries teeth were primary incisors of the upper jaw maxilla, indicating a dominance of the ECC type II [55]. We explain our lower evidence of ECC II and III with the necessity of an earlier treatment need of children with special health care needs, with an early indication for dental treatment under GA due to various factors (medication, etc.) compared with the healthy children.

Regarding the limitations of our study, the key point is that only a single-center study for children with special healthcare needs was conducted and reported. Confirmation of our results with an enrollment of a larger number of children and a longer follow-up would be desirable. Furthermore, due to the retrospective study design, no calibration of pre-treatment was feasible. Thus, prospective study designs are recommended for further investigations.

## 5. Conclusions

Within the limitations of this retrospective study, data exhibited high caries experience in all patient groups regardless of age and coexisting factors with need for dental treatment. A high waiting time in all groups demonstrates the importance of increasing opportunities for dental care for these special children, who have to be treated under GA and, moreover, the need for prevention in these high-risk groups.

## Figures and Tables

**Figure 1 jcm-11-02613-f001:**
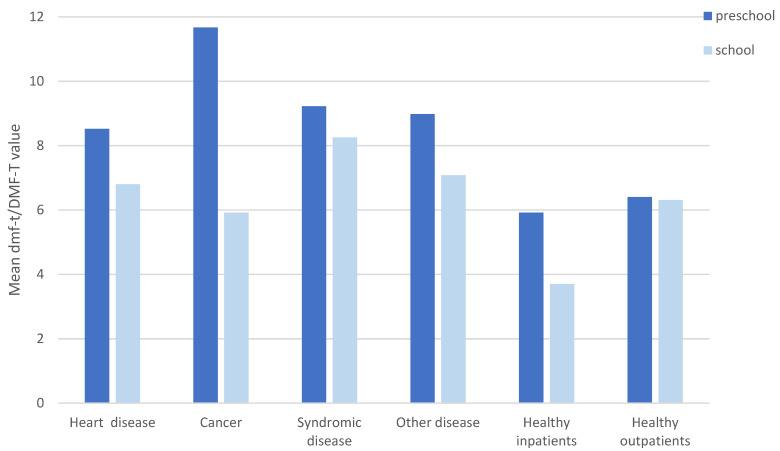
Mean dmf-t/DMF-T values in the different groups at impulse 1.

**Figure 2 jcm-11-02613-f002:**
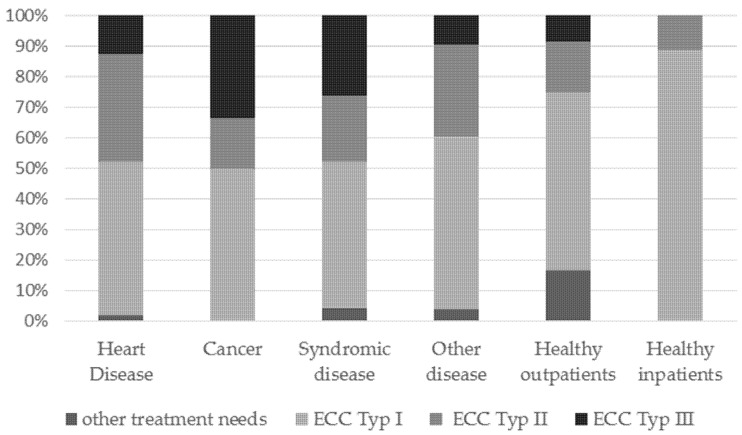
Distribution of ECC according to Wyne [38] regarding each group of treated children at baseline.

**Figure 3 jcm-11-02613-f003:**
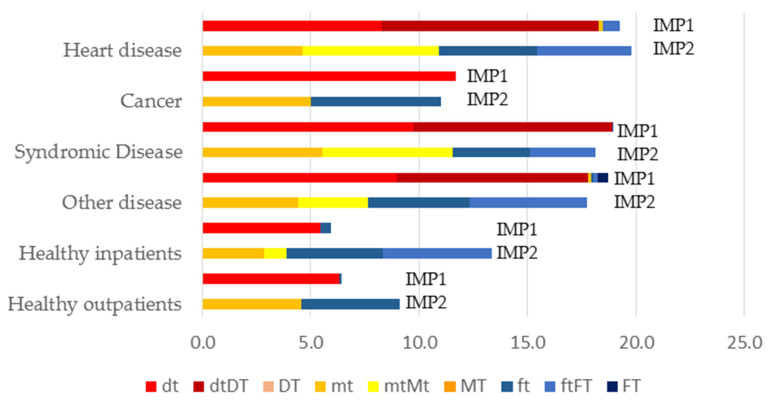
dmf-t/DMF-T index of pre-school children at first impulse (IMP1-pre-operative assessment) and second impulse (IMP2-dental treatment under GA): decayed (dt), missing (mt), and filled (ft) teeth for sole primary dentitions; decayed (dtDT), missing (mtMT), and filled teeth (ftFT) for mixed dentitions; decayed (DT), missing (MT), and filled (FT) teeth for sole permanent dentitions.

**Figure 4 jcm-11-02613-f004:**
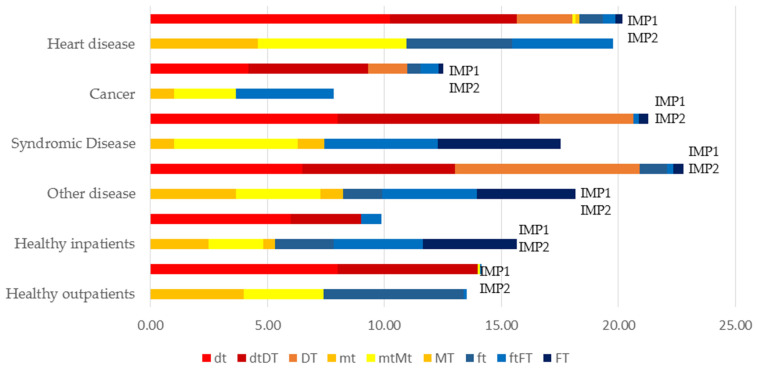
dmf-t/DMF-T index of school children at first impulse (IMP1-pre-operative assessment) and at second impulse (IMP2-dental treatment under GA): decayed (dt), missing (mt), and filled (ft) teeth for sole primary dentitions; decayed (dtDT), missing (mtMT), and filled teeth (ftFT) for mixed dentitions; decayed (DT), missing (MT), and filled (FT) teeth for sole permanent dentitions.

**Table 1 jcm-11-02613-t001:** Treatment need of all groups regarding Significant Caries Index (SiC) and Caries Restoration Index (CRI) for impulse 1.

Pre-Existing Cofactors	SiC Index		CRI Index [%]	
	Pre-School Children	School Children	Pre-School Children	School Children
Heart disease	13.81 ± 2.67 (n = 46)	11.30 ± 4.17 (n = 30)	5.20	4.96
Cancer	17.00 ± 2.00 (n = 6)	10.25 ± 1.92 (n = 12)	0.00	12.22
Syndromic disease	14.29 ± 1.83 (n = 22)	13.46 ± 3.75 (n = 32)	0.53	41.08
Other disease	14.59 ± 2.55 (n = 52)	11.29 ± 2.73 (n = 63)	2.67	8.17
Healthy inpatients	11.50 ± 3.04 (n = 24)	9.00 ± 0.82 (n = 10)	8.47	8.90
Healthy outpatients	11.80 ± 1.83 (n = 15)	9.00 ± 1.23 (n = 13)	1.56	7.06

**Table 2 jcm-11-02613-t002:** Waiting time (in months) for dental treatment under general anesthesia according to group between impulses 1 and 2.

Dentition	Pre-Existing Cofactor	Pre-School Children			School Children		
		Mean [mo]	SD [mo]	n	Mean [mo]	SD [mo]	n
Sole Primary Dentition	Heart disease	6.3	4.5	37	6.7	3.6	6
	Cancer	3.5	3.9	6	1.0	1.0	3
	Syndromic disease	6.3	4.8	14	45.5	58.7	2
	Other disease	4.9	4.2	42	8.9	16.4	9
	Healthy inpatients	3.4	2.7	20	26	38.1	3
	Healthy outpatients	1.2	2.8	15	1.0	/	1
Mixed Dentition	Heart disease	6.3	1.5	3	5.0	5.4	23
	Cancer	/	/	/	3.5	3.7	8
	Syndromic disease	6.0	1.7	3	5.4	6.2	27
	Other disease	2.5	2.1	2	6.3	5.6	52
	Healthy inpatients	15.0	19.8	2	7.9	13.4	7
	Healthy outpatients	/	/	/	0.6	0.5	11
Sole Permanent Dentition	Heart disease	/	/	/	26.8	36.2	7
	Cancer	/	/	/	36.0	/	/
	Syndromic disease	/	/	/	17.1	19.9	8
	Other disease	/	/	/	14.3	11.9	10
	Healthy inpatients	/	/	/	22.0	22.6	2
	Healthy outpatients	/	/	/	0	/	1

**Table 3 jcm-11-02613-t003:** Number of dental procedures in GA patients (lower case words for primary and capitalized words for permanent teeth).

	Heart Disease	Cancer	Syndromic Disease	Other Disease	Healthy Inpatients	Healthy Outpatients
fillings	241	54	136	304	101	167
ssc	8	6	12	33	6	31
extraction	296	55	233	403	31	42
Fillings	54	18	96	166	26	15
SSC	-	-	-	-	-	-
Extraction	18	5	20	29	1	-

## Data Availability

Not applicable.

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
