# Peer review of "Dental Treatment under General Anesthesia in Pre-School Children and Schoolchildren with Special Healthcare Needs: A Comparative Retrospective Study"

_jcm, 2022, doi:10.3390/jcm11092613_

Round 1

Reviewer 1 Report

This manuscript describes the evaluation of dental treatments received under general anesthesia in preschool children and schoolchildren with special healthcare needs. In children with severe health conditions oral health frequently overlooked in spite of the importance it has for the quality of life and even systemic health of the child. Therefore the issue is relevant and pertinent.

However the study design, the hypothesis stated, the comparisons presented in the results and discussed are confusing and there doesn´t seem to be logical link between hypothesis, comparisons and conclusion. 

For example the statement in line 45-49 leaves the reader confused as to what the study is about. Is it going to relate caries experience with preexisting cofactors (meaning preexisting systemic disease), which is implicit in the first sentence. Or is it going to compare the dental treatments done under general anesthesia in preschool and school children with special heath care needs as describes in the second sentence? Reading the null hypothesis it seems to aim to achieve both, but the so  impulse 1 and 2  and the preschool vs school comparisons described in the methods and results make it very confusing to understand. 

The introduction, although supported by citations, lacks an overall logic sequence which leads the reader to the issue focused by the authors. 

Materials and methods are also not clear. For example it is not clear what is considered an in or outpatient and therefore the need to include private practice patients. Furthermore it is not clear if the methodology used in the observations (regarding caries index scoring) and data collection is standardized between these 2 collection sites. The purpose of 2 times for the assessment of the caries indexes (impulse 1 and 2) is also not clear. What is tested by this comparison which is not included in the hypothesis.

The group description in lines 122- 128 is also very confusing. children without general disease are the children in the healthy group? Why not call them that? 

Regarding the results it is not always clear as to how they support the testing of the hypothesis. For example Figure one has the comparison between preschool and school children at impulse 1 which is not part of the hypothesis to be tested. What is the relevance of the Figure? Another example is waiting time (Table 2) although it is certainly an important factor, it is not clear as to how it relates to what the authors propose to test with the study. 

 Finally the conclusion seems completely out of scopus. I can not find any support for it neither in the results nor the discussion. Furthermore it is not related to the hypothesis proposed. 

Sentence in line 61-62 is not clear and must be improved.

Statement in lines 102 and 103 seem impossible to do: dental treatment needs are not necessarily dental treatments provided, so you can not "evaluate dental treatment needs recieved under GA". Please clarify.

Lines  105-106 refer to methods and seem out of place. 

Sentence in line 173 is strange. What is meant by exploratory data analysis? 

Reviewer 2 Report

This article is very clear structured and gives a good overview of the topic .

Comments:

line 98: instead of" radical" path --> "extensive "

line 141: further information concerning the questionnaire and its content are helpful for better background understanding

description of figure 3 is not right. It is actually showing the status of the dmft indices and not the performed therapy

check the form of citation, especially in the discussion part in case of multiple citations (beginning with the smallest number and then in ascending order)

Round 2

Reviewer 1 Report

Revisions made in the definition of the objectives, the added paragraph in the discussion and the rewriting of the conclusion, greatly improve the readibility of the manuscript. I recommend only minor text editing:

  • line 346 "highlights" instead of "highlight"
  • lines 385,386 "data show a high caries experi-
    ence in all patient groups regardless of  age" instead of "data exhibited high caries experi-
    ence in all patient groups indifferent from age"  

Author Response

Dear Reviewer,

Thank you for your comments, which were well received.

According to your suggestions, the following changes have been made.

Best regards

Comments and Suggestions for Authors

Revisions made in the definition of the objectives, the added paragraph in the discussion and the rewriting of the conclusion, greatly improve the readibility of the manuscript. I recommend only minor text editing:

  • English language and style

( ) Extensive editing of English language and style required
(x) Moderate English changes required
( ) English language and style are fine/minor spell check required
( ) I don't feel qualified to judge about the English language and style

Our response: Thank you for your comment. Before submission, a native English speaker edited our manuscript. Please see the English Editing Certificate attached. We checked the manuscript again for English improvement.

Revised text: See English Editing Certificate.

  • line 346 "highlights" instead of "highlight"

Our response: Thank you for your comment. We corrected the wording.

Revised text: See page 10, line 346.

  • lines 385,386 "data show a high caries experience in all patient groups regardless of  age" instead of "data exhibited high caries experience in all patient groups indifferent from age"  

Our Response: Thank you for your comment. We corrected the wording.

Revised text: See introduction section, page 11, line 386.